

# Early warning model construction and validation for urinary tract infection in patients with neurogenic lower urinary tract dysfunction (NLUTD): a retrospective study

Liqiong Zhou[1,2,*], Surui Liang[1,*], Qin Shuai[3], Chunhua Fan[3], Linghong Gao[3] and Wenzhi Cai[1]

[1] Nursing Department, Southern Medical University, Shenzhen Hospital, Shenzhen, Guangdong, China
[2] Nursing Department, Southern Medical University, Shenzhen, Guangdong, China
[3] Nursing Department, Shenzhen Longcheng Hospital, Shenzhen, Guangdong, China
* These authors contributed equally to this work.

## ABSTRACT

**Background**. This study was performed to construct and validate an early risk warning model of urinary tract infection in patients with neurogenic lower urinary tract dysfunction (NLUTD).

**Methods**. Eligible patients with NLUTD admitted to Shenzhen Longcheng hospital from January 2017 to June 2021 were recruited for model construction, internal validation and external validation. The first time point of data collection was within half a month of patients first diagnosed with NLUTD. The second time point was at the 6-month follow-up. The early warning model was constructed by logistic regression. The model prediction effects were validated using the area under the Receiver Operating Characteristic curve, the Boostrap experiment and the calibration plot of the combined data. The model was externally validated using sensitivity, specificity and accuracy.

**Results**. Six predictors were identified in the model, namely patients ≥65 years old (OR = 2.478, 95%CI [1.215– 5.050]), female (OR = 2.552, 95%CI [1.286–5.065]), diabetes (OR = 2.364, 95%CI) [1.182–4.731]), combined with urinary calculi (OR = 2.948, 95%CI [1.387–6.265]), indwelling catheterization (OR = 1.988, 95%CI [1.003 – 3.940]) and bladder behavior training intervention time ≥2 weeks (OR = 2.489, 95%CI [1.233–5.022]); and the early warning model formula was $Y = 0.907 \times$ age+ 0.937 × sex + 0.860 × diabetes +1.081 × combined with urinary calculi+ 0.687 × indwelling catheterization+ 0.912 × bladder behavior training intervention time-2.570. The results show that the area under the ROC curve is 0.832, which is close to that of 1,000 Bootstrap internal validation (0.828). The calibration plot shows that the early warning model has good discrimination ability and consistency. The external validation shows the sensitivity is 62.5%, the specificity is 100%, and the accuracy is 90%.

**Conclusion**. The early warning model for urinary tract infection in patients with NLUTD is suitable for clinical practice, which can provide targeted guidance for the evaluation of urinary tract infection in patients with NLUTD.

Corresponding author
Wenzhi Cai,
caiwenzhi2002@hotmail.com,
caiwzh@smu.edu.cn

# INTRODUCTION

Neurogenic lower urinary tract dysfunction (NLUTD) refers to abnormal function of either the bladder, bladder neck, and/or its sphincters caused by lesions to the central or peripheral nervous system (*Jahromi, Mure & Gomez, 2014*; *Panicker, DeSèze & Fowler, 2013*). It is often clinically manifested as poor urination, urinary retention, abnormal ejaculation and other symptoms such as lower urinary tract and sexual dysfunction (*Amarenco et al., 2017*). NLUTD could cause a series of urinary complications, such as hydronephrosis, pyelonephritis and urinary tract infection (UTI) (*Lima et al., 2015*; *Manack et al., 2011*). Significantly, UTI is the most common complication. *Unsal-Delialioglu et al., (2010)* reported that 70% of febrile spinal cord injury patients with NLUTD in rehabilitation units developed urinary tract infections. Meanwhile, another study also shows the prevalence of UTI is>50% among patients with NLUTD (*Nicolle et al., 2005*). Once a patient develops UTI, it will seriously affect the rehabilitation of the primary disease and may further cause secondary damage to the upper urinary tract, leading to renal function damage and even death (*Amarenco et al., 2017*; *Wagenlehner & Pilatz, 2018*). Furthermore, UTI is an independent risk factor leading to renal damage, and UTI has serious adverse effects on the prognosis of patients with NLUTD (*Chow, Gallo & Busse, 2018*).

As for effective preventive measures, intermittent catheterization may be associated with fewer urinary tract infections (*Hooton et al., 2010*; *Siroky, 2002*). The evidence showed that bladder irrigations, the use of hydrophilic catheters and the frequency of intermittent self-catheterization can reduce the risk of UTI (*Cox et al., 2017*; *Lapides et al., 1972*; *Vo et al., 2013*). However, there is no consensus among current studies on the effects of prophylactic antibiotics for preventing UTI. Catheters impregnated with antibiotics or silver-coated catheters have shown short-term effects on bacteriuria and infection (*Prieto et al., 2017*; *Salameh, Mohajer & Darouiche, 2015*). However, with prolonged use, antibiotic resistance may develop (*Salameh, Mohajer & Darouiche, 2015*). In addition, aseptic bacteriuria should not be treated because treatment of asymptomatic bacteriuria does not prevent febrile urinary tract infection or infection recurrence and increases bacterial resistance and virulence due to antibiotics (*D'Hondt & Everaert, 2011*; *Vasudeva & Madersbacher, 2014*). Bacterial interference may promise to prevent UTI, but more evidence is currently needed (*Dinh et al., 2019*).

Given the high incidence and severe impacts of UTI among NLUTD patients, early identification of risk factors for UTI in patients with NLUTD is essential. Common risk factors of UTI in patients with NLUTD include diabetes mellitus, hypoalbuminemia, long-term indwelling catheter, multiple intermittent catheterization, frequent bladder irrigation and intervention time for bladder function retraining (*Fang, Shubin & Huaping, 2018*; *Kinnear et al., 2020*; *Pang & Li, 2013*; *Zhang, Hua & Li, 2021*). However, identifying clinical risk factors for UTI is based on experts' opinions, without using contemporary

modelling techniques, which is not helpful in accurately classifying patients. Furthermore, some researchers have shown that independent risk factors for UTI in patients with NLUTD include indwelling catheter, bladder capacity <200 ml, increased bladder pressure and bladder emptying method (*Anderson et al., 2019*; *Zhang, Hua & Li, 2021*). However, no study has investigated whether bladder treatment-related parameters are predictors for urinary tract infection in patients with NLUTD. Therefore, this study collected general information, disease characteristics and bladder treatment-related parameters of patients with NLUTD to construct the early warning UTI model among NLUTD patients. This study could provide novel insights for clinical early screening of such patients, which could strengthen the prediction of UTI in patients with NLUTD during rehabilitation and adopt active individuals for patients with different risk levels.

## MATERIALS AND METHODS

### Participants

Eligible patients with NLUTD admitted to Shenzhen Longcheng Hospital from December 2017 to December 2020 were retrospectively selected as construction and internal validation research subjects, and patients from January 2021 to June 2021 were selected as external validation research objects. The inclusion criterion was patients first diagnosed with NLUTD (ICD-10 code N31.901), while the exclusion criteria were (1) UTI co-existing with pulmonary infection, wound infection, etc., namely secondary UTI; (2) cannot obtain the required medical records, such as lack of blood routine, renal function, urological imaging data or urodynamic; (3) patients did not complete the 6-month follow-up.

### Data collection

We collected the following predictors half-month after patients diagnosed with NLUTD: (1) General clinical data: including gender, age, body mass index (BMI), diabetes, hypertension, urinary calculi, and so on (2) Disease characteristics and bladder treatment-related data: including NLUTD type based on clinical manifestation (detrusor overactivity type, detrusor inactivity type and mixed type), catheterization method (voiding/intermittent catheterization and indwelling catheterization), preventive use of antibacterial drugs, time to start the bladder behavior training, serum albumin, serum creatinine, total serum bilirubin and blood urea nitrogen. Time to start the bladder behavior training <2 weeks means that the training started within two weeks of the first diagnosis of NLUTD. The bladder behaviour training included scheduled urination, delayed urination, reflex urination, anal stretch techniques, and pelvic floor muscle training. Patients with indwelling catheterization only performed anal stretch techniques and pelvic floor muscle training.

The end point outcome observation time is at six-month follow-up. All patients underwent routine urinalysis and urine culture tests. The clinical diagnosis of UTI met the relevant standards in the "Guidelines on urological infections" formulated by the European Association of Urology (*EAU Guidelines, 2022*). Clinical symptoms of UTI include urgent urinary, frequent urination, urinary retention, dysuria, lumbago, and urination pain. Physical signs included fever, costovertebral angle tenderness/pain and suprapubic tenderness. Patients with an axillary temperature of >37.5 were considered
febrile. According to the above clinical manifestations, this study assesses UTI with "yes" if one of the following conditions is met and with one of above clinical manifestations: (1) Diagnosis of UTI in medical records; (2) Positive urine culture; (3) Fresh urine specimens were centrifuged and examined by phase contrast microscopy (400 ×). Bacteria were observed in half of every 30 visual fields.

### Ethical approval

This study was approved by the Medical Ethics Committee of Shenzhen Longcheng Hospital (LCLL(3)–2021). Since this study is a retrospective study and there is no medical risk to the patients, the patient's informed consent was exempted with the approval of the Medical Ethics Committee.

### Statistical analysis

The data was analyzed by SPSS 22.0 software and R statistics version 3.6.3, utilizing the ROCR package for all analyses. The count data of patients' age, BMI and diabetes were described by frequency and percentage. The comparison between UTI and non-UTI groups was performed by univariate regression analysis. $P < 0.05$ indicated that the difference was statistically significant. The logistic regression model was used to construct the risk prediction model. The prediction ability of the model was evaluated by the AUROC (Are Under Receiver Operating Characteristic) curve, C-index and the calibration curve. Bootstrapping with 1,000 resamples was done to calculate a relatively accurate C-index for internal cross-validation (*Steyerberg, 2019*; *Van Calster et al., 2016*; *Van Calster & Vickers, 2015*). The external validation was conducted by using sensitivity, specificity and accuracy (*Steyerberg, 2019*).

## RESULTS

There were 206 patients for the model construction and internal validation (See Fig. 1). The age ranged from 35 to 88, and their mean age was 62.79 ± 18.95. Among them, 60 (29.1%) had UTI, while 146 patients did not develop UTI. There were 30 patients for the external validation, 19 males and 11 females. The age ranged from 20 to 86 (51.72 ± 13.51).

### Univariate regression analysis of NLUTD patients between groups

The clinical data of patients with UTI and those without UTI was analyzed by univariate analysis. The results showed that the proportion of ≥65 years old, complicated with diabetes, urinary calculi, hypoproteinemia and female patients in the UTI group was significantly higher than that of the non-UTI group. The proportion of patients with intermittent catheterization and time to start bladder training <2 weeks was significantly lower in the UTI group than in the non-UTI group ($P < 0.05$). See Table 1.

### Multivariate regression analysis among NLUTD patients with UTI

The statistically significant factors in the univariate analysis were used as independent variables, and whether the patient had a UTI during the follow-up period was used as the dependent variable, and binary logistic regression analysis was performed (see Table 2).

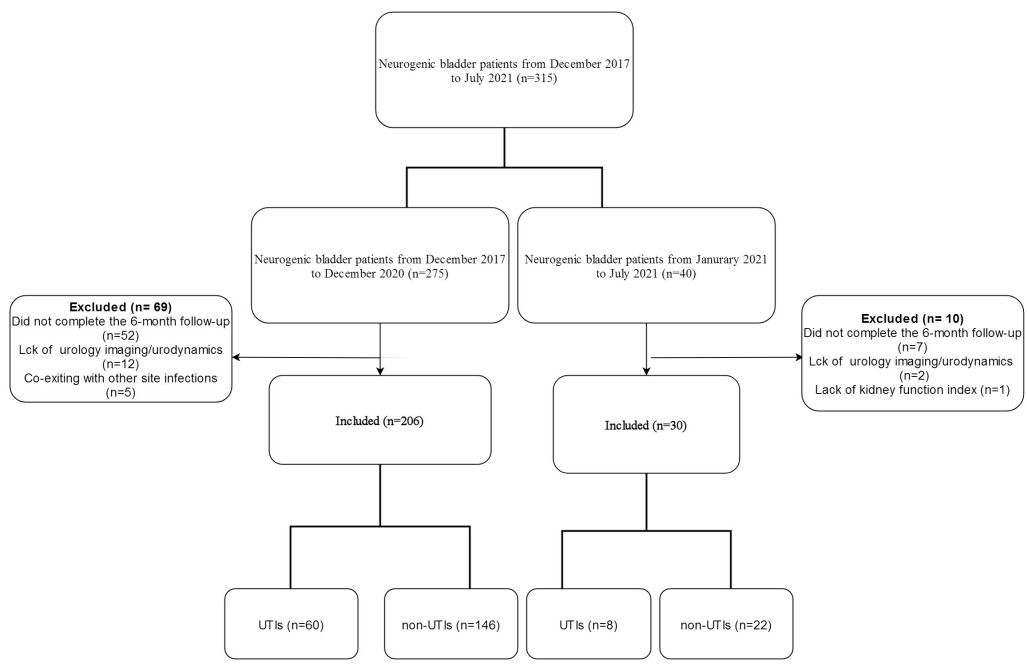

**Figure 1  Flowchart of inclusion.**

The results showed that under the premise of controlling for other confounding factors, patients age $\geq 65$ (OR $= 2.478$, 95%CI [1.215–5.050]), female patients (OR $= 2.552$, 95%CI [1.286–5.065]), diabetes (OR $= 2.364$, 95% CI [1.182–4.731]), combined urinary calculi (OR $= 2.948$, 95% CI [1.387–6.265]), indwelling catheterization embodiment (OR $= 1.988$, 95% CI [1.003–3.940]) and bladder behavior training intervention time $\geq 2$ weeks (OR $= 2.489$, 95%CI [1.233–5.022]) were independent risk factors for UTI in patients with NLUTD ($P < 0.05$). The final formula was Y $= 0.907\times$ age $+ 0.937\times$ female gender $+ 0.860\times$ diabetic $+ 1.081\times$ urinary calculi $+ 0.687\times$ indwelling catheterization $+ 0.912\times$ Time to start bladder behavior training $\geq 2$ weeks $+ 0.547\times$ hypoproteinemia $-2.570$. See Table 2 for details. Y means the calculated UTI risk prediction score.

### Internal evaluation of the early warning model

The AUROC of the model was evaluated, and the initial C-index was 0.832 (see Fig. 2). The Youden index was $0.82 + 0.59 - 1 = 0.41$. The C-index of 1,000 Bootstrap internal validations was 0.828 (see Fig. 3). The calibration curve showed that the predicted value of the model and the observed value had a certain correlation (see Fig. 4), indicating that the model had a higher probability of predicting the occurrence of UTI and the actual probability of occurrence.

### External validation of the early warning model

Thirty eligible patients with NLUTD treated in Shenzhen Longcheng hospital from January to June 2021 were included for external validation. There were 19 males and 11 females. The age ranged from 20 to 86 ($51.72 \pm 13.51$). According to the formula of this prediction

**Table 1** Univariate analysis among NLUTD patients with or without UTI ($n = 206$).

| Factor | UTI group ($N = 60$) | Non-UTI group ($N = 146$) | x2 | P |
|---|---|---|---|---|
| Age | | | 6.898 | 0.009 |
| ≥65 | 42 (70.0) | 73 (50.0) | | |
| <65 | 18 (30.0) | 73 (50.0) | | |
| Gender | | | 6.967 | 0.008 |
| Male | 28 (46.7) | 97 (66.4) | | |
| Female | 32 (53.3) | 49 (33.6) | | |
| BMI (kg/m2) | | | 0.833 | 0.361 |
| ≥24 | 22 (36.7) | 44 (30.1) | | |
| <24 | 38 (63.3) | 102 (69.9) | | |
| Hypertension | | | 2.513 | 0.113 |
| Yes | 26 (43.3) | 81 (55.5) | | |
| No | 34 (56.7) | 65 (44.5) | | |
| Diabetes | | | 6.668 | 0.010 |
| Yes | 29 (48.3) | 43 (29.5) | | |
| No | 31 (51.7) | 103 (70.5) | | |
| Urinary calculi | | | 7.688 | 0.006 |
| Yes | 23 (38.3) | 29 (19.9) | | |
| No | 37 (61.7) | 117 (80.1) | | |
| NLUTD type | | | 1.142 | 0.565 |
| Detrusor overactivity type | 5 (8.3) | 18 (12.3) | | |
| Detrusor inactivity type | 40 (66.7) | 99 (67.8) | | |
| Mixed type | 15 (25.0) | 29 (19.9) | | |
| Catheterization method | | | 6.154 | 0.013 |
| Intermittent catheterization | 31 (51.7) | 102 (69.9) | | |
| Indwelling catheterization | 29 (48.3) | 44 (30.1) | | |
| Preventive use of antibacterial drugs | | | 2.598 | 0.107 |
| Yes | 39 (65.0) | 77 (52.7) | | |
| No | 21 (35.0) | 69 (47.3) | | |
| Time to start the bladder behavior training (weeks) | | | 6.467 | 0.011 |
| <2 weeks | 22 (36.7) | 82 (56.2) | | |
| ≥2 weeks | 38 (63.3) | 64 (43.8) | | |
| Hypoproteinemia | | | 4.875 | 0.027 |
| Yes | 17 (28.3) | 22 (15.1) | | |
| No | 43 (71.7) | 124 (84.9) | | |
| Serum creatinine | | | 2.184 | 0.139 |
| Normal | 45 (75.0) | 94 (64.4) | | |
| Abnormal | 15 (25.0) | 52 (35.6) | | |
| Serum total bilirubin | | | 1.303 | 0.254 |
| Normal | 53 (88.3) | 136 (93.2) | | |
| Abnormal | 7 (11.7) | 10 (6.8) | | |

**Table 1** (*continued*)

| Factor | UTI group (N = 60) | Non-UTI group (N = 146) | x2 | P |
|---|---|---|---|---|
| Diagnosis | | | 1.099 | 0.577 |
| Spinal cord injury | 42 (71.3) | 107 (73.5) | | |
| Cerebrovascular accident/ brain tumor/ stroke | 12 (19.6) | 21 (14.4) | | |
| Peripheral neuropathy (such as diabetes) | 6 (10.1%) | 18 (12.1) | | |

**Notes.**
BMI, body mass index; NLUTD, neurogenic lower urinary tract dysfunction; UTI, urinary tract infection.

model, when calculated UTI prediction score ≥ 0.41, it is considered that the patient will occur UTI. This model predicts that 5 patients will occur UTI and 25 patients will not occur UTI within 6-month follow-up. The results were that UTI occurred in 8 patients and non-UTI occurred in 22 patients. Compared with the actual results, the sensitivity of the prediction model is 62.5%, the specificity is 100%, and the accuracy is 90%.

## DISCUSSION

The risk factors of UTI in patients with NLUTD are complex, which is the result of the multiple influencing factors. The occurrence is related to the localized bacterial colonization, due to patient's decreased ability to urinate autonomously and the loss of urine flushing in the urethra, and also associated with the patients' own physiological and pathological factors, as welas treatment methods. In this study, we found that age ≥65 years old, female patients, combined with diabetes, urinary calculi, indwelling catheterization and bladder behavior training intervention time ≥ 2 weeks could be used to identify the high risk of UTI in patients with NLUTD. Sixty patients occurred UTI in 206 patients with NLUTD; the incidence rate was 29.1%, significantly lower than the 45.37% (*Shi, Bai & Chen, 2020*) and 45.59% (*Pang & Li, 2013*) reported by the other domestic researchers. The main reason may be the difference of the NLUTD pathogenic factors between this study and other studies. In the analysis of *Shi, Bai & Chen (2020)* and *Pang & Li (2013)*, most reported research objects are mainly NLUTD patients caused by elderly cerebrovascular accidents. Such patients often have a poor prognosis, and their bedtime is significantly longer, which leads to a substantially higher risk of UTI. In the study, 70% of patients had NLUTD caused by spinal cord injury, while about 20% patients had cerebrovascular accidents, stroke, or brain tumor. Therefore, the incidence of UTI in this study was significantly lower than in other reports.

### The research significance of the early warning model

UTI is one of the common complications of NLUTD. Because of a variety of complex factors, it can often manifest as recurrent episodes, which not only aggravates the clinical symptoms of NLUTD, but also affects the renal parenchyma damage, and ultimately lead to a decline in the quality of life of patients and shortened life expectancy (*Van Calster & Vickers, 2015*). Therefore, it must be actively prevented and controlled.

Identification of clinical risk factors for UTI is previously based on experts' opinions without using contemporary modelling techniques, which is not helpful to accurately

Zhou et al. (2022), *PeerJ*, DOI 10.7717/peerj.13388

**Table 2  Multivariable regression analysis among NLUTD patients ($n = 206$).**

| Factor | Assignment method | Regression coefficients | Standard error | Waldvalue | P | OR | 95%confidence interval | |
|---|---|---|---|---|---|---|---|---|
| | | | | | | | Upper limit | Lower limit |
| Age | ≥65years old = 1, <65years old = 0 | 0.907 | 0.363 | 6.235 | 0.013 | 2.478 | 1.215 | 5.050 |
| Gender | Female = 1, Male = 0 | 0.937 | 0.350 | 7.173 | 0.007 | 2.552 | 1.286 | 5.065 |
| With diabetes | Yes = 1, No = 0 | 0.860 | 0.354 | 5.912 | 0.015 | 2.364 | 1.182 | 4.731 |
| Combined with urinary calculi | Yes = 1, No = 0 | 1.081 | 0.385 | 7.898 | 0.005 | 2.948 | 1.387 | 6.265 |
| Catheterization method | Indwelling catheterization = 1, Intermittent catheterization = 0 | 0.687 | 0.349 | 3.870 | 0.049 | 1.988 | 1.003 | 3.940 |
| Time to start bladder behavior training | ≥2 weeks = 1, <2 weeks = 0 | 0.912 | 0.358 | 6.476 | 0.011 | 2.489 | 1.233 | 5.022 |
| Hypoproteinemia | Yes = 1, No = 0 | 0.547 | 0.422 | 1.674 | 0.196 | 1.727 | 0.755 | 3.952 |
| constant | | −2.570 | 0.638 | 16.223 | 0.000 | 0.077 | | |

**Notes.**

NLUTD, neurogenic lower urinary tract dysfunction.

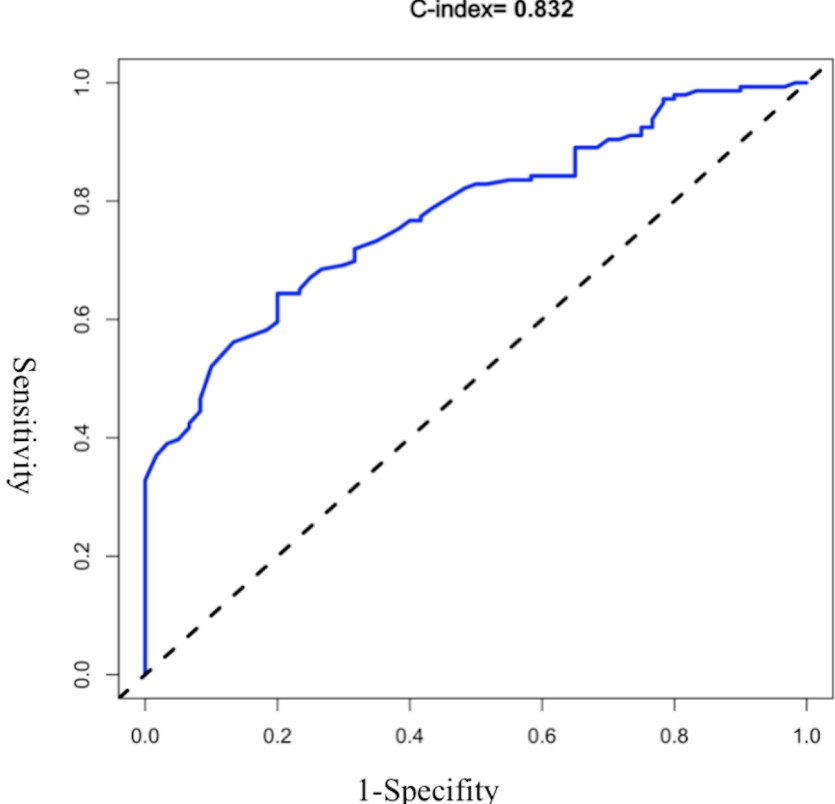

**Figure 2  ROC curve of the initial early warning model.**

classify patients, leading to insufficient prevention and treatment of UTI (*Chow, Gallo & Busse, 2018*). The establishment of the predictive model shifts the focus from interventions when UTI occurs to the purposeful and focused prevention of UTI in people with high-risk NLUTD. This early warning model intuitively displays the relevant risk factors and their correlation with the occurrence of UTI, which could arouse the attention of medical staff, improve prevention, and finally reduce the workload of medical staff.

## The prediction effect of the early warning model

In this study, the ROC curve was used to test the model's fit, with specificity as the abscissa and sensitivity as the ordinate. The area under the ROC curve obtained in this study is 0.832, and the area under the ROC curve verified internally is 0.828. The optimal value of the area under the ROC curve is 1. Moreover, the results of the calibration chart in this study are better, and its slope is basically the same as the standard curve. The external validation results also showed a specificity of 100% and an accuracy of 90%.

## Elderly female patients are prone to UTI

Elderly patients are prone to infections due to decreased estrogen levels, broad and short urinary tract, poor nutritional status, and reduced immunity. Studies have shown that with age, the probability of occurrence of UTI increases significantly because female patients

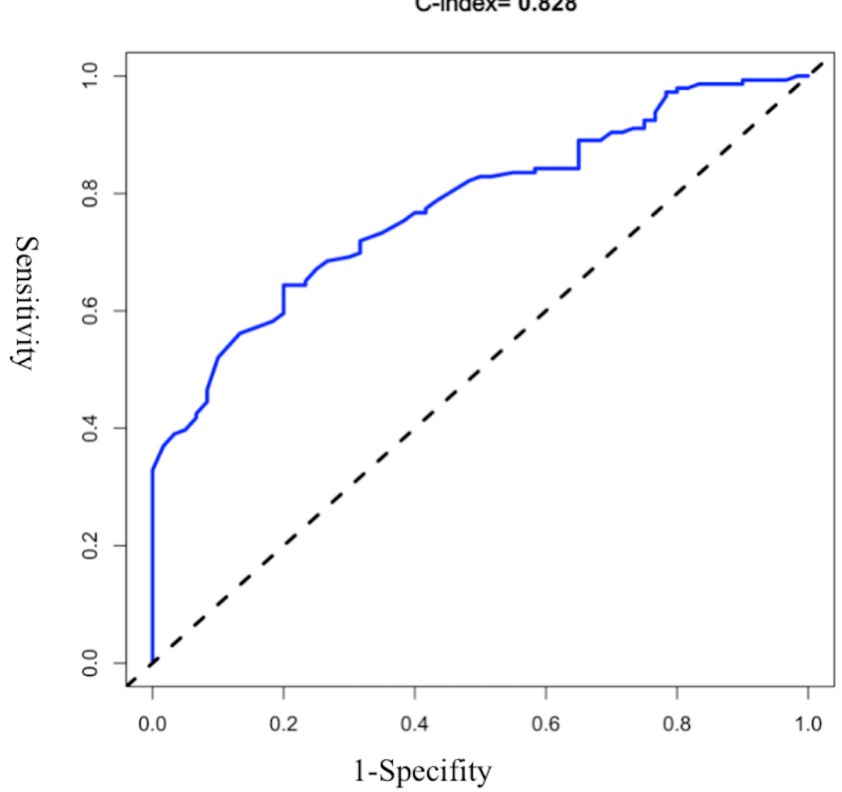

C-index= 0.828

**Figure 3** ROC curve of 1,000 bootstrap internal validations.

have a broad and short urinary tract, which is close to the anus, leading to bacterial colonization and retrograde infection (*Kakde, Redkar & Yelale, 2018*). Especially in elderly menopausal female patients, due to the decline of ovarian function, the level of estrogen decreases significantly, which eventually leads to the degeneration of the reproductive system, a decrease in blood supply to the urinary tract, and a reduction in muscle tone that maintains the bladder and urethra (*Jung & Brubaker, 2019*).

## Patients with diabetes and urinary calculi are prone to UTI

Many studies have confirmed that diabetes is closely related to UTI, and the incidence of UTI has positively correlated with the course of diabetes, blood sugar levels and glycosylated haemoglobin (*He et al., 2018*; *Zubair et al., 2019*). The reduction of substance metabolism in diabetic patients reduces the body's synthesis of immunoglobulins, antibodies and complement, thus minimizing the chemotaxis, phagocytosis and sterilization of inflammatory cytokines. Therefore, the body's cellular immune barrier is destroyed, causing the drop of the immunity system. In addition, patients with a long course of diabetes may be accompanied by peripheral nerve and vascular disease, which may cause urinary tract blood supply disorders and lead to local neurological abnormalities, degrading the genitourinary system and causing infection (*Hine et al., 2017*). Studies have shown that urinary calculi are closely related to the occurrence of UTI, and patients with urinary calculi

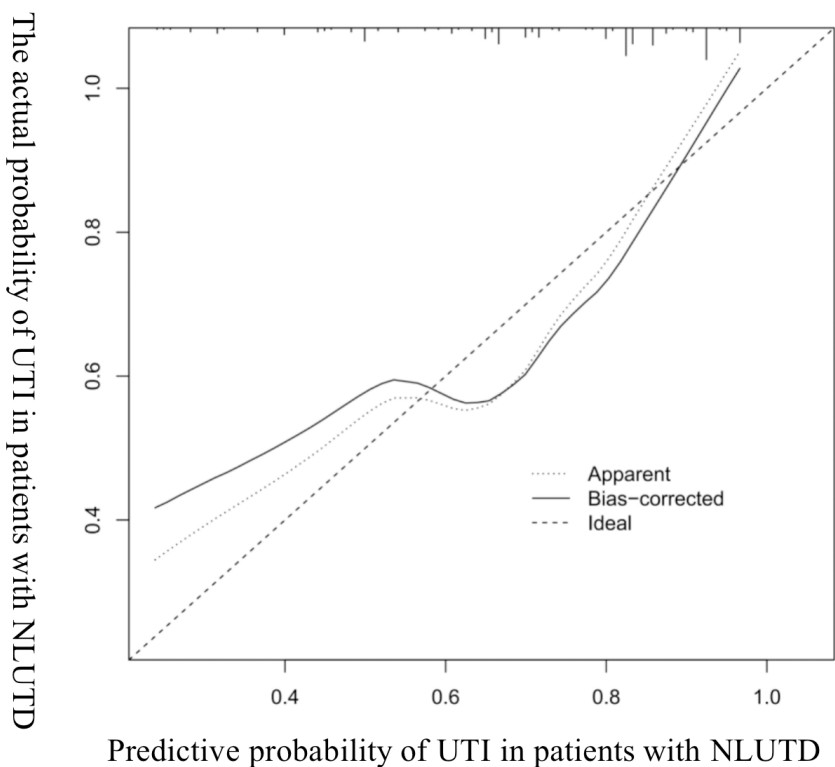

**Figure 4 Calibration curve diagram of UTI in patients with NLUTD.**

(diameter of 2 cm or more) are at higher risk of recurrent UTI (*Wang & Li, 2020*). The main reason is that bacteria tend to colonize on the irregular surface of urinary calculi, which prevents the antibacterial drugs from reaching the site of action, resulting in poor effects of antibacterial treatment.

## Patients who use indwelling catheterization are prone to UTI

Intermittent urinary catheterization is strongly recommended for patients with NLUTD, which is suitable for reducing the incidence of UTI. Still, the indwelling catheterization method had to be used for some severely ill patients (extreme weakness or ureteral reflux patients) (*Han et al., 2017*). However, as a foreign body, the urinary catheter can cause reactive inflammation when it comes in contact with the urethral mucosa and bladder mucosa. The urethral mucosa is damaged during the catheter operation, which further reduces the urethra physical barrier function and may cause infection after bacteria enter the urethra retrogradely from the urethral orifice to outside the official cavity (*Menegueti et al., 2019*). For patients with indwelling catheterization, patients should use the closed drainage catheter system (*Nicolle et al., 2005*), and indwelling catheters should not be replaced systematically but in the event of an obstruction, hematuria, or UTI (*Hooton et al., 2010*). UTI may be reduced by early anal stretch techniques and pelvic floor muscle training, which could help patientsimprove bladder function recovery, and reduce bladder residual urine output (*Ginsberg, 2013*).

### The correlation between hypoproteinemia and UTI remains to be verified

The study showed that the decrease of serum albumin levels in patients with NLUTD is related to UTI. The reason may be the decreased immunity of patients with hypoalbuminemia (*Jiao, 2020*). In this study, univariate analysis showed statistical significance between groups ($P < 0.05$). However, regression analysis showed no statistical difference between hypoproteinemia and UTI. The main reason for this difference may be that hypoalbuminemia in this study mainly occurred in elderly patients. The hypoalbuminemia may be excluded as a confounding factor of this model. In addition, It may also be due to the small sample size of patients with hypoproteinemia ($n = 39$). Future studies are needed to explore whether hypoproteinemia could be used to predict UTI among patients with NLUTD.

### Strengths and limitations of this research

This study attempts to develop and validate an early warning model of UTI among patients with NLUTD compared with previous studies. This model has the characteristics of simple data acquisition and quick evaluation, which can provide purposeful and focused prevention of UTI in people with NLUTD (*Autumn et al., 2018*). The limitation of this study lies in that it was only verified in one hospital. This study suggests that it still needs to be verified by a large sample, multi-centre, and prospective study before the model is routinely applied to pre-clinical applications. To develop a model that was easy to use in clinical settings, we dichotomized the continuous age variable into a binary variable. This may lose more information on the variable itself.

## CONCLUSION

The predictive effect of this early warning model for UTI in patients with NLUTD is suitable for clinical practice, which can provide targeted guidance for the early prediction of UTI in patients with NLUTD.

### Funding

This study was supported by the Science and Technology Project of Shenzhen (No. JCYJ20210324142406016). The funders had no role in study design, data collection and analysis, decision to publish, or preparation of the manuscript.

### Grant Disclosures

The following grant information was disclosed by the authors:
Technology Project of Shenzhen: JCYJ20210324142406016.

### Competing Interests

The authors declare there are no competing interests.

## Author Contributions

- Liqiong Zhou conceived and designed the experiments, performed the experiments, analyzed the data, prepared figures and/or tables, authored or reviewed drafts of the paper, and approved the final draft.
- Surui Liang conceived and designed the experiments, analyzed the data, prepared figures and/or tables, authored or reviewed drafts of the paper, and approved the final draft.
- Qin Shuai, Chunhua Fan and Linghong Gao conceived and designed the experiments, performed the experiments, authored or reviewed drafts of the paper, and approved the final draft.
- Wenzhi Cai conceived and designed the experiments, authored or reviewed drafts of the paper, and approved the final draft.

## Human Ethics

The following information was supplied relating to ethical approvals (i.e., approving body and any reference numbers):

The Medical Ethics Committee of Shenzhen Longcheng Hospital approved this study (LCLL(3)–2021).

## Data Availability

The raw data of all measurements are available in the Supplementary File.

## Supplemental Information

Supplemental information for this article can be found online at http://dx.doi.org/10.7717/peerj.13388#supplemental-information.

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
