# Peer review of "Early warning model construction and validation for urinary tract infection in patients with neurogenic lower urinary tract dysfunction (NLUTD): a retrospective study"

_PeerJ, doi:10.7717/peerj.13388_

## Round 0.1 · original submission · Major Revisions

Please pay careful attention to all major concerns raised by reviewer 1.

·

Basic reporting

Thank you for the possibility to review this manuscript. I am especially pleased that the document comes from a nursing department, as nurses and nurse practitioners are usually more involved in patient care and the crucial link for a successful patient management. The authors construct and validate an early warning model for UTI susceptibility in patients with neurogenic lower urinary tract dysfunction. While such a model is highly warranted there are several concerns I have several concerns. One main risk factor (i.e. recurrent UTIs / previous history of UTIs) is not included in the model, the patients population is not very well described and while the study describes important risk factors the results would not allow any statements regarding targeted guidance or early intervention.

Major concerns:
(1) Please delete all conclusions regarding targeted guidance or possibilities for early intervention. The provided manuscript does not allow any assumptions in this direction as it does not focus on intervention or on prophylactic measures but on risk factor assessment.
(2) The most common risk factor for UTI is previous UTIs in the patient history. Why was this not included in the analyses. Could the model be corrected for this risk factor?
(3) I strongly suggest to adhere to the to the terminology provided by the standardization committees from the International Continence Society (e.g. Abrams et al. 2002, NAU, The standardization of terminology of lower urinary tract function and Gajewski et al. 2018, NAU, An International Continence Society report on the terminology for adult neurogenic lower urinary tract dysfunction (ANLUTD)). Terms like neurogenic bladder, even if often used, are misleading as the functional problem is not limited to the bladder but to the bladder, the sphincter and the urethra (i.e. neurogenic bladder = neurogenic lower urinary tract dysfunction). Other not correctly used terms can be found in Table 1 / lines 89-90: While I would understand the term “Detrusor hyperreflexia” (what would should be “Detrusor overactivity”) I do not know what the following terms mean: “Detrusor non-reflex type and Hybrid”. There are several other terms that need to be corrected.
(4) Line 81-84: The exclusion criteria should be refined: “(1) Patients with infections from other parts of the body, or UTI followed by non-primary infectious lesions; (2) Patients with asymptomatic bacteriuria; (3) UTI caused by non-NGB; (4) Patients with severe heart disease dysfunction, liver and kidney dysfunction, blood system disease or malignant tumor disease, etc.”
• (1) What does “UTI followed by non-primary infectious lesions” means in this contest?
• (2) For a risk model at least two timepoint for each patient are needed, hence, it would not make sense to exclude patients if they showed asymptomatic bacteriuria at one but a UTI at another visit, please clarify
• (3) UTIs are not caused by the bladder, please rephrase
• (4) I would not understand why patients with heart disease would need to be excluded, please explain.
(5) Lines 92-94: The authors make the statement: “The clinical diagnosis of UTI met the relevant standards in the "Guidelines on urological infections" formulated by the European Association of Urology (European Association, 2021)” However, up to date there is no definition for UTI in patients with neurogenic lower urinary tract dysfunction, while there are definitions for uncomplicated UTI, complicated UTI and specific sub-types such as catheter associated UTI. Please explain the exact UTI definition used for this manuscript (what symptoms were regarded UTI? What symptoms were not regarded UTI? Would it be possible to provide a table with the frequency of the different symptoms in the ?)?, how many patients had febrile UTI, how many urosepsis? etc.). In addition, please check the following link and use the format suggested for citing the EAU Guidelines https://uroweb.org/guidelines/how-to-cite-the-eau-guidelines/
(6) At least two patient visits are needed for such risk models. Please explain how many timepoints per patient were used. Please show in the results the time differences between the visits for both groups (UTI and non-UTI) and explain how you corrected for different intervals in the model.
(7) It would be interesting to get more information on patient characteristics. While some information can be drawn from table 1, it would further be important to have insights on following information: (1) Main underlying neurological diagnosis (2) in-patients vs out-patients (3) days/years after diagnosis of the neurogenic lower urinary tract dysfunction
(8) Please explain bladder behavior training intervention in the methods in more details. In the discussion following information is given (lines 221-223): “bladder behavior training (scheduled urination, delayed urination, mental urination, and anal stretch techniques)” however, regarding table 1 all subjects in the study relayed on some type of catheterization to empty the bladder.
(9) Lines 164-165: In the discussion the authors make following statement: “In the study, 70% of patients had NGB caused by lumbar disc disease, however in table 1 it seems that all patients included in the analyses relied on some type of catheterization for bladder emptying. Please describe in more detail why so many patients with “lumbar disc diseases” could not void spontaneously, I would not expect that “lumbar disc disease” would cause such a high number of neurogenic lower urinary tract dysfunction / need for catheterization, unless it is a cauda equina syndrome or a conus-cauda syndrome.
(10) In the discussion the authors make following statement (line 177): “focused prevention of UTI in people with high-risk NGB”, in the introduction following statement is made (Lines 48-51): Studies have shown that the incidence of UTI in patients with NGB was up to 70% despite the use of individualized urinary catheterization or preventive use of antibacterial drugs (Amarenco et al., 2017). This seems to be contradictory, please clarify what preventive measures should be taken in what patients.
(11) In patients performing intermittent self-catheterization there is evidence that bladder irrigations, the use of hydrophilic catheters and the frequency of intermittent self-catheterization can reduce the rist of UTI (Lapides et al., J Urol, 1972; Husmann Transl Adnrol Urol, 2016; Cox et al., Can Urol Assoc J, 2017; Li et al., Phys Med Rehabil, 2013). Please discuss.

Minor concerns
(1) Lines 55-57: A recent study has shown that the incidence of chronic renal failure in patients with NGB complicated by UTI was about 24% (Na et al., 2014). Please rephrase or delete in this sentence, in the current version it could be seen misleading.
(2) Line 58-62: please add history of UTI to the list.
(3) Line 190: please rephrase the term: “decreased visceral function”
(4) The ethical vote is written in the Chinese language. Please translate the relevant parts in English.

Experimental design

Please see "major concerns" in the basic reporting section:
(2); (4); (5); (6);

Validity of the findings

Please see "major concerns" in the basic reporting section:
(1); (10); (11)

Additional comments

Please see "major concerns" in the basic reporting section:
(3); (7); (8); (9) and all minor concerns.

Reviewer 2 ·

Basic reporting

1. Please cite the statistical software in line 101, standards in line 93,

2. In line 104, the logistic regression "model" instead of "equation"?

3. In line 132, please define Z.

4. There is no need to include Table 2 since it has redundant information with Table 3. The information in Table 2 can be easily incorporated into Table 3 by adding the levels to the 1st column.

Experimental design

1. In Line 77, please provide more details on the number of participants screened, number of patients meeting inclusion criteria and etc. It would be great if a flow diagram of enrollment is provided.

2. In Table 1, why is age modeled as a binary variable instead of a continuous variable? Dichtomizing continuous variables tend to lose more information on the variable itself.

Validity of the findings

1. In line 181-187, the authors commented that the prediction effect is medium and good since the ROC value of 0.832 was evaluated in the training dataset. However, we don't make comments on performance using the internal ROC due to the overfitting issue on training data.

2. In line 231, the authors mentioned that analysis's multicollinearity of hypoalbuminemia and age excluded hypoalbuminemia as a founding factor. Please properly evaluate the multicollinearity among these variables to back up this claim.

---

## Round 0.2 · accepted · Accept

The authors satisfactorily resolved the reviewers' concerns.

Reviewer 2 ·

Basic reporting

no comment

Experimental design

no comment

Validity of the findings

no comment

Additional comments

The authors have addressed all my comments.